# Sampling, Filtering, and Analysis Protocols to Detect Black Carbon, Organic Carbon, and Total Carbon in Seasonal Surface Snow in an Urban Background and Arctic Finland (>60° N)

**Outi Meinander \*, Enna Heikkinen, Minna Aurela**  **and Antti Hyvärinen**

Finnish Meteorological Institute, Erik Palmeninaukio 1, 00560 Helsinki, Finland; enna.heikkinen@fmi.fi (E.H.); minna.aurela@fmi.fi (M.A.); antti.hyvarinen@fmi.fi (A.H.)
\* Correspondence: outi.meinander@fmi.fi

**Abstract:** Black carbon (BC), organic carbon (OC), and total carbon (TC) in snow are important for their climatic and cryospheric effects. They are also part of the global carbon cycle. Atmospheric black and organic carbon (including brown carbon) may deposit and darken snow surfaces. Currently, there are no standardized methods for sampling, filtering, and analysis protocols to detect carbon in snow. Here, we describe our current methods and protocols to detect carbon in seasonal snow using the OCEC thermal optical method, a European standard for atmospheric elemental carbon (EC). We analyzed snow collected within and around the urban background SMEARIII (Station for Measuring Ecosystem-Atmosphere Relations) at Kumpula (60° N) and the Arctic GAW (Global Atmospheric Watch) station at Sodankylä (67° N). The median BC, OC, and TC in snow samples ($n_{tot}$ = 30) in Kumpula were 1118, 5279, and 6396 ppb, and in Sodankylä, they were 19, 1751, and 629 ppb. Laboratory experiments showed that error due to carbon attached to a sampling bag (n = 11) was <0.01%. Sonication slightly increased the measured EC, while wetting the filter or filtering the wrong side up indicated a possible sample loss. Finally, we discuss the benefits and drawbacks of OCEC to detect carbon in snow.

**Keywords:** snow; seasonal; black carbon; organic carbon; brown carbon; carbon; urban background; Arctic; protocol; procedure

---

## 1. Introduction

Atmospheric black carbon (BC) originates mainly from incomplete combustion of carbonaceous materials, for example fossil fuels and biomass. Black carbon is defined [1] as a distinct type of carbonaceous material that is formed primarily in flames, is directly emitted to the atmosphere, and has a unique combination of physical properties: it strongly absorbs visible light, is refractory with a vaporization temperature near 4000 K, exists as an aggregate of small spheres, and is insoluble in water and common organic solvents. Organic carbon (OC), in turn, is co-emitted with black carbon or may come from local soils [1] and refers to carbon mass that is not black. Here, OC is the quantity that results from thermal analysis of carbon aerosols and enables the reporting of the elemental carbon to organic carbon (EC-to-OC) ratio. Detailed scientific assessments of the climatic and cryospheric effects of BC have been presented in [1–3]. Various impacts of BC deposited in snow and ice have been widely investigated in, e.g., [4–14]. It is estimated that a large fraction of particulate light absorption in Arctic snow (about 30–50%) is due to non-BC constituents, especially light-absorbing organic carbon, including so-called brown carbon, which originates from biofuel and agricultural or boreal forest burning [1]. A higher OC/EC ratio in snow (205:1) than air (10:1) was found in [15], suggesting that

snow could be influenced by water soluble gas phase compounds. Light absorption of the Barrow snowpacks, in turn, was suggested to be due to humic like substances (HULIS), which are part of brown carbon [16]. HULIS optical properties were found to be consistent with aged biomass burning or a possible marine source [17].

Currently, there are no standardized methods for sampling, filtering, and analysis protocols to detect atmospheric carbon deposited in snow. Earlier, the protocols for BC content in snow for seasonal Svalbard snowpacks on glaciers were reported [18]. The Svalbard protocols were designed in order to exclude the complexity of soil–snow interactions and to focus instead on snow–atmosphere interactions. In addition, the snowpack chemistry monitoring protocol for the Rocky Mountain Network was given in [19]. Here, we describe in detail our current methods for sampling, filtering, and analysis protocols to detect black, organic, and total carbon (TC being the sum of BC and OC) in seasonal surface snow in an urban background environment and in the Finnish Arctic, north of 60º N. For this study, elemental carbon is used synonymously with black carbon due to their measurement technique dependence in snow impurity studies. The focus of this paper is on atmosphere–snow interactions, i.e., atmospheric carbon wet and dry deposited in snow.

## 2. Materials and Methods

### 2.1. Site Description

For this study, we analyzed snow samples ($n_{tot}$ = 30) collected within and around SMEARIII (Station for Measuring Ecosystem-Atmosphere Relations) and the Finnish Meteorological Institute (FMI) located in Kumpula, Helsinki (60° N), and the Finnish Meteorological Institute's GAW (Global Atmospheric Watch) station of Sodankylä-Pallas (67° N) (Figure 1). We also studied carbon attached to the sampling bags (n = 11, Section 2.6).

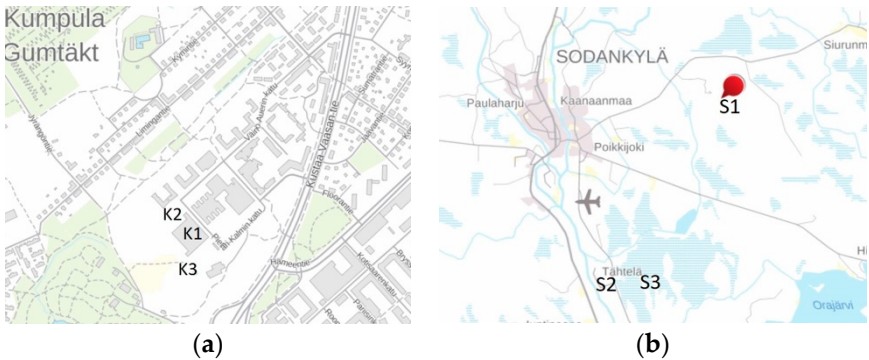

(**a**)                 (**b**)

**Figure 1.** (**a**) The sampling sites at Kumpula (K1 is the roof of the Finnish Meteorological Institute FMI; K2 is the fenced land area of FMI; K3 is Station for Measuring Ecosystem-Atmosphere Relations SMEARIII). (**b**) The sampling sites at Sodankylä (S1 is Kommattivaara; S2 is the Nordkalotten Satellite Evaluation co-operation Network NorSEN mast at Tähtelä; S3 is the site next to the SnowAPP field). The maps have a north-up orientation. (National Land Survey open data Attribution CC 4.0 license).

Kumpula is located 5 km northeast of the Helsinki city center and represents an urban background. The SMEARIII station [20] is operated together with the University of Helsinki and the Finnish Meteorological Institute. The Sodankylä-Pallas GAW station of the Arctic Space Centre in Sodankylä is located at 67.37° N, 26.63° E, in the boreal forest zone. The station area consists of open areas of mineral soil and coniferous forests, as well as an open peat bog [21]. Seasonal snow cover in Sodankylä belongs to the taiga class [22]. The station is part of the World Meteorological Organization (WMO) Global Cryosphere Watch (GCW) network.

Snow sampling at Kumpula represented the SMEARIII area and was made at the measurement roof and in the fenced land area of the Finnish Meteorological Institute. For Sodankylä, samples collected during the SNORTEX (Snow Reflectance Transition Experiment) 2009–2010 [23] at Kommattivaara

and the NorSEN (Nordkalotten Satellite Evaluation co-operation Network) Mast were used, and snow surface samples were collected next to the fenced SnowAPP (Modelling of the Snow microphysical-radiative interaction and its APPlications) field [24]. These samples were also used for laboratory experiments to quantify some of the error sources.

## 2.2. Snow Sampling and Sample Storage

Here, we sampled surface snow, which we define as the top 2–3 cm of the surface snow within an approximately 1 m$^2$ sampling area, and the amount of collected snow varied from deciliters to liters. The large volume samples were divided for the experimental part of the work to determine some of the error sources. More details can be found in the Supplementary Materials.

In Kumpula and Sodankylä, we have freezers specifically for snow samples, and the samples are easily stored there after sampling. In the sites, we did not use gloves in the field, but we carefully avoided touching the sample or the inside of the sampling bag by hand. We used clean stainless-steel tools, and sterile plastic bags, except for samples identified as "non-sterile bags." Non-sterile bags were used because we ran out of sterile bags, and the impact of using these non-sterile bags on the carbon analysis results is a subject included in this study.

Good practices in field work for the chemical analysis of carbon in seasonal snow include making notes on GPS coordinates, date and local time of sampling, taking photos of the site, and writing notes on environmental and meteorological conditions, such as strong wind, snow fall, and anything that might affect the snow conditions. To avoid contamination, snow sampling is best performed using clean stainless steel, glass, or sterile plastics materials in all sampling. Snow samples can be collected in sterile plastic bags and containers or glass containers to avoid contamination. Snow needs to be sampled using a sterile tool without touching the snow sample or the inside of the sampling container with the hands. Clean gloves are recommended. Field blank filters are kept in order to detect and estimate any possible contamination. Sampling can be performed according to the snow depth or snowpack layers, each having their benefits. One option is to collect surface snow and/or the whole snowpack depth. After sampling, snow samples are transported frozen and are kept frozen until melted for filtering. If the sample starts to melt, it should be filtered as soon as possible.

How much snow is needed for one carbon analysis depends, on the one hand, on how dirty the snow is in order to avoid carbon oversaturation in the OCEC analysis and, on the other hand, on how clean it is in order to obtain a sample above the carbon detection limit. Therefore, the best practice is to prepare and analyze a set of test snow samples to characterize the circumstances prior to any larger campaign to gain a priori knowledge on the snow carbon contents. If snow is collected in multiple bags, a good practice is to put one spatula of snow after another in different bags (e.g., one spatula of snow in bag number one, the next spatula in bag number two, and the third in bag number one). The frequency of sampling depends on the application and will be further addressed in Section 4, Discussion.

## 2.3. Melting and Filtering

The snow samples of Kumpula and Sodankylä were taken one at a time from the freezer to be melted and filtered. The frozen snow sample was divided into two parts, and both parts were put into beakers and weighed. Both samples were melted in a microwave oven at melting temperature, which took approximately 15 min. Immediately after melting, one of the samples was filtered through the pre-burned (800 °C, 4 h) quartz filter (diameter 47 mm, model T293, Munktell Filter AB, Falun, Sweden; T293 is not pre-burned, while MK360 is the same filter sold as pre-burned). The beaker was washed with 3 × 30 mL ultrapure Milli-Q water (Millipore Corporation, Burlington, MA, USA) and the water was filtered together with the sample. The other sample was sonicated in an ultrasonic bath for 10 min immediately before filtering it in the same way.

The principle for the snow sample melt for filtering (Supplementary Figures S1–S6) is to melt it slowly and with low heat in order to avoid evaporation. The filtered melt water amount is needed for concentration conversion. Most often, the melt is done in laboratory conditions. Visible trash is

taken away using a clean, stainless steel sieve. A clean laboratory microwave reserved especially for the purpose can be used, or melt can take place in a laboratory at an indoor temperature inside the sampling bag or container. It is necessary to ensure that no dust or impurities be deposited from the air to the sample during the melt. After melting the sample, the snow is filtered through sterile quartz filters for the thermal-optical analysis. Quartz filters are used because they sustain temperatures over 900 °C, which they will be subjected to in thermal-optical analysis. The filter is placed with the hairy side up on top of a conical flask using a clean pair of tweezers.

## 2.4. OCEC Analysis

Here, a total of 41 samples (30 snow samples and 11 bag rinsing samples) were analyzed using a thermal-optical carbon analyzer (OCEC, Sunset Laboratory, Tigard, OR, USA, model 5 L). A rectangular piece (1 × 1.5 cm) was cut from the sample filter with a cutting tool for analysis (Figure S7). The filter piece was put on the glass sample spoon of the OCEC instrument using tweezers, and the sample spoon was lifted into the sample oven. The samples were analyzed using the EUSAAR2 protocol. A blank and a sucrose standard was run every day before running any samples. The volume of meltwater was used for concentration conversions into parts per billion by mass (ppb), which is equal to μg/kg, determined as μg-EC/L-$H_2O$ in the OCEC method. The uncertainty of the OCEC method for lightly loaded samples was estimated to be +/−0.2 μgC (+5% relative error for higher loaded samples, according to Sunset Laboratory Inc.).

The thermal-optical OCEC method used in this study is the current European standard for determining atmospheric EC [25,26]. The method was created by [27], where a more detailed description of OCEC is given. The basic principle of OCEC is as follows: A filter piece is put inside the sample oven and heated in two phases so that all the carbon on the filter is oxidized to carbon dioxide ($CO_2$). In the first phase, helium atmosphere is used to oxidize OC into $CO_2$. In the second phase, an oxygen-helium mixture is used to vaporize and oxidize the EC into $CO_2$. Carbon dioxide is reduced to methane, which is measured using a flame ionizing detector (FID). The split point, which separates OC and EC, is determined using laser transmittance because, during the first phase, some of the OC is pyrolized, and it is released during the second phase. During pyrolysis, the laser transmittance signal decreases. At the split point, the laser transmittance signal returns to its original value.

## 2.5. Analysis Protocols

An OCEC analysis protocol determines the number, temperature, and length of temperature steps in the heating program. The most commonly used protocols are EUSAAR 2 (EUropean Supersites for Atmospheric Aerosol Research), NIOSH 870 (National Institute of Occupational Safety and Health), IMPROVE (Interagency Monitoring of Protected Visual Environments), and IMPROVE A (Table 1). All the protocols yield very similar values for total carbon. However, the fraction of organic carbon to elemental carbon varies between protocols. In this study, the EUSAAR2 protocol was used.

**Table 1.** Various protocols used in thermal optical OCEC analysis and their parameters, EUSAAR 2 (EUropean Supersites for Atmospheric Aerosol Research) and IMPROVE (Interagency Monitoring of Protected Visual Environments).

| | EUSAAR 2 | | IMPROVE A | |
|---|---|---|---|---|
| **Atmosphere** | **Temperature (°C)** | **Time (s)** | **Temperature (°C)** | **Time (s)** |
| He | 200 | 120 | 140 | 150–580 |
| He | 300 | 150 | 280 | 150–580 |
| He | 450 | 180 | 480 | 150–580 |
| He | 650 | 180 | 580 | 150–580 |
| He | 0 | 30 | - | - |
| He/$O_2$ | 500 | 120 | 580 | 150–580 |
| He/$O_2$ | 550 | 120 | 740 | 150–580 |
| He/$O_2$ | 700 | 70 | 840 | 150–580 |
| He/$O_2$ | 850 | 80 | - | - |
| He/$O_2$ | - | - | - | - |
| He/$O_2$ | - | - | - | - |

## 2.6. Laboratory Experiments

To identify and quantify some of the error sources, several experiments were performed using the Kumpula and Sodankylä snow samples. These experiments included the following:

(1)　errors due to carbon remaining in the sampling bags when removing the sample (i.e., not rinsing the sample bag);
(2)　errors due to carbon remaining in the melt container (i.e., not rinsing the melting container);
(3)　errors due to filter leaking or wetting (i.e., large amounts of water passing through the filter);
(4)　errors using the sample and/or reference filter with the upper side down, filtering, and measuring.

For (1), we used two types of plastic bags to store the snow samples, namely sterile polyethylene (Nasco Sampling LLC/Whirl-Pak, www.enasco.com, Fort Atkinson, WI, USA) and non-sterile deep-freeze polyethylene plastic bags (Minigrip, Amerplast Ltd., www.amerplast.com, Tampere, Finland), to estimate how sensitive the snow samples were for storage material. When comparing the differences in filter loadings, we used OCEC analysis and absorption/attenuation measurements with a particle soot absorption photometer, PSAP [28], at wavelengths of 467 nm (blue), 530 nm (green), and 660 nm (red) [29]. The basis for the optical attenuation measurements is the exponential attenuation of light as it passes through some medium, often described by the Bouguer–Lambert–Beer law (Equation (1)) as follows:

$$I = I_0 e^{-\tau} \tag{1}$$

where $I_0$ is the light intensity through a clean reference filter, and $I$ is the light intensity through a sample loaded filter. The exponent $\tau$ is the optical depth of impurities on the filter.

### 2.6.1. Bag Rinsing

In order to see if there was any significant sample loss due to some of the black carbon adhering to the walls of the sample bag, all the bags were rinsed with 100 mL of Milli-Q water after the snow had been removed (Supplementary Figure S8). The Milli-Q water was poured into the bag, and the bag was shaken to remove any black carbon. The water was then filtered through a quartz filter.

### 2.6.2. Sonication

Snow samples (Supplementary Table S1) were taken one at a time out of the freezer. The snow sample was divided into four or two parts depending on its size before melting. All samples except two were divided into four parts, two of which were sonicated before filtering (Supplementary Figure S9). Two samples were divided into two parts as there was not enough snow for four parts. Of the two

parts, one was sonicated before filtering. Two of the sub-samples were melted at the same time in a microwave oven at low heat using a melting program for about 15 min. One of the sub-samples was filtered immediately after melting, while the other was sonicated in an ultrasound bath for 10 min and filtered immediately after.

### 2.6.3. Filter Wetting

A base solution was made by taking a small amount of wood-burning soot and adding 1000 mL of Milli-Q water. The base solution was used to make seven diluted solutions with a concentration of 1/10 the base solution (Table S2). Different volumes of the diluted solution were filtered through quartz filters to test whether any filter leaking or wetting was observed when large amounts of water passed through the filter. The filters were dried overnight in a laminar cabinet. A circular 0.64 cm$^2$ piece was taken from each filter, and PSAP was used to measure filter transmittance.

### 2.6.4. Filter Wrong Side up

A 1/10 diluted solution was made from the BC base solution, as described in the previous section, and 200 mL of the diluted solution was filtered through a quartz filter that had been placed wrong side up on the filter holder (Figure 2). The filter was dried, and the transmittance measured with PSAP. Several PSAP measurements (n = 4) were made while changing the orientation of the sample filter piece as well as the orientation of the blank filter. Similar measurements were made using a filter where 200 mL of the diluted solution had been filtered on the right side of the filter.

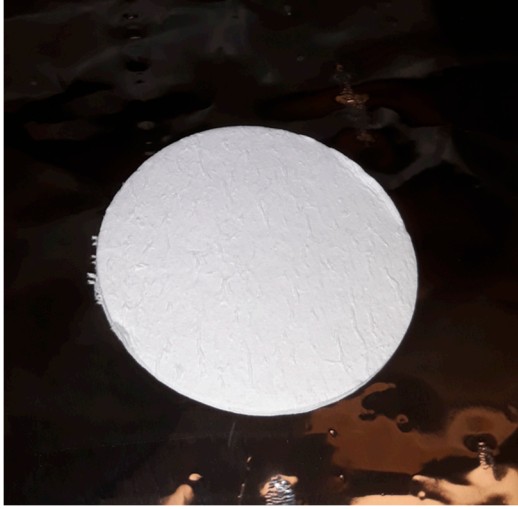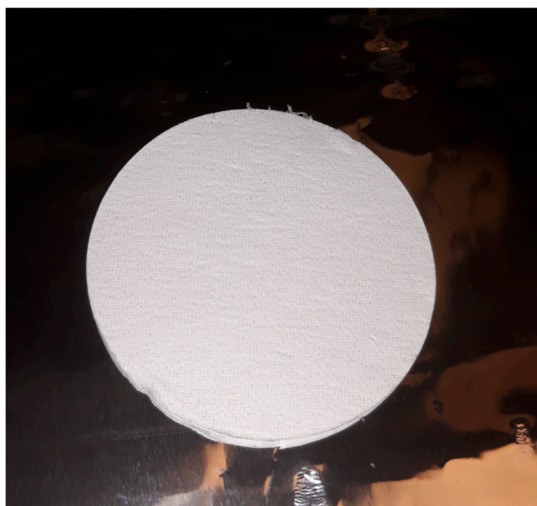

**Figure 2.** A filter right side up (**left** picture) and wrong side up (**right** picture).

## 3. Results

### 3.1. Carbon in Snow

The median black carbon (determined as elemental carbon), organic carbon, and total carbon in snow samples in Kumpula were 1118, 5279, and 6396 ppb, and in Sodankylä, they were 19, 1751, and 629 ppb, respectively (Table 2). The number of snow samples analyzed with OCEC was n = 30 (28 samples presented in Figure 3, and two SnowAPP snow samples collected at a meter-scale distance from each other). Hence, the total number of snow carbon results was 90 (3 × 30). The median OC/EC ratio in the snow samples was 8:1 in Kumpula and 140:1 in Sodankylä.

**Table 2.** Snow sample results of organic (OC), elemental (EC), and total carbon (TC). Number of snow samples analyzed with OCEC $n_{tot}$ = 30. The uncertainty of the OCEC is estimated to be +/−0.2 µgC (+5% relative error for higher loaded samples).

| Sample ID | Place | Sampling Day | OC [ppb] | EC [ppb] | TC [ppb} |
|---|---|---|---|---|---|
| TOC (AoS-2015 experiment) | Kumpula Dynamicum fenced area | 6 February 2015 | 1838.4 | 164.3 | 2002.7 |
| Kumpula Roof | Kumpula Dynamicum Measurement Roof | 24 February 2014 | 8719.4 | 2071.3 | 10,790.7 |
| NorSEN 1 L | Sodankylä NorSEN mast | 23 March 2010 | 608.6 | 21.9 | 630.5 |
| NorSEN 1S | Sodankylä NorSEN Mast | 23 March 2010 | 3046.6 | 18.8 | 3065.4 |
| NorSEN M | Sodankylä NorSEN Mast | 23 March 2010 | 423.2 | 6.7 | 429.8 |
| KOI5 | Sodankylä Kommattivaara | 23 March 2010 | 2873.9 | 20.6 | 2894.5 |
| SnowAPP | Sodankylä site next to SnowAPP field | 18 March 2019 | 627.7 | 1.5 | 629.2 |

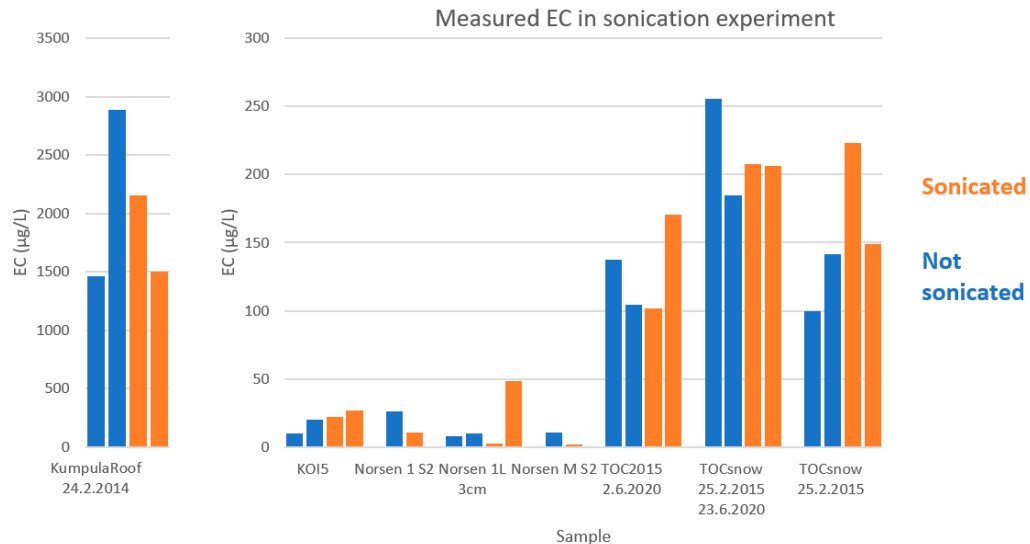

**Figure 3.** The elemental carbon (EC) results of the sonication experiment in a pairwise comparison of the sonicated and non-sonicated samples.

*3.2. Rinsing Sampling Bags*

Most rinsing samples left no elemental carbon, or extremely small amounts of it, in the bag after snow was removed (Supplementary Figure S10). Only two samples had elemental carbon above the detection limit of the OCEC instrument [µgC]. These were Pallasbag3 (4.71 µgC) and KumpulaRoof (2.02 µgC). This shows that some of the elemental carbon may adhere to the walls of the container bag. However, when compared to the total amount of elemental carbon in the snow sample, the amount left on the bag is miniscule (less than 0.01%). The material of the sample bag did not seem to affect the amount of elemental carbon left on the walls of the sample bag, as both sterile bags (Whirl-Pak) and non-sterile bags (Minigrip) had some rinsing samples with zero elemental carbon and others with detectable amounts of elemental carbon.

### 3.3. Sonication

The measured EC values varied quite substantially (Figure 3). There were large differences in EC values not only between sonicated and non-sonicated sub-samples but also between all the sub-samples from the same snow sample. This may be partly due to the original snow sample being heterogeneous in EC distribution since it was divided as snow into sub-samples. Another reason might be the OCEC measurement. For some of the sub-samples, the split point was near the top of a larger signal peak.

In the sonication experiment, the sub-snow samples (Figure 3) differed pairwise (sonicated sub sample minus non-sonicated) for Kumpula from −1381 to 695 ppb EC and for Sodankylä from 4 to 38 ppb EC, while the standard deviations varied for Kumpula from 5 to 977, and for Sodankylä, from 4 to 27, respectively. Figure 3 shows the variability within one snow sample (one or two pairs of sonicated/non-sonicated cases per sample) and the effect of sonication. Sonication seemed to have slightly increased the measured EC amount, indicating the possibility of using sonication to extract EC attached to the container walls. This difference, however, is small compared to the difference caused by the snow heterogeneity and OCEC measurement.

### 3.4. Filter Wetting

The filter wetting results were expected to show a linear decrease of transmittance when increasing the volume of the filtered solution (Figure 4). With this solution, this was true until the filtered solution amount exceeded 300 mL. Thereafter, the decrease was smaller. This slightly suggests that some of the soot might not stay in the filter, but instead, the filter would start leaking. Further studies are needed to confirm this preliminary finding.

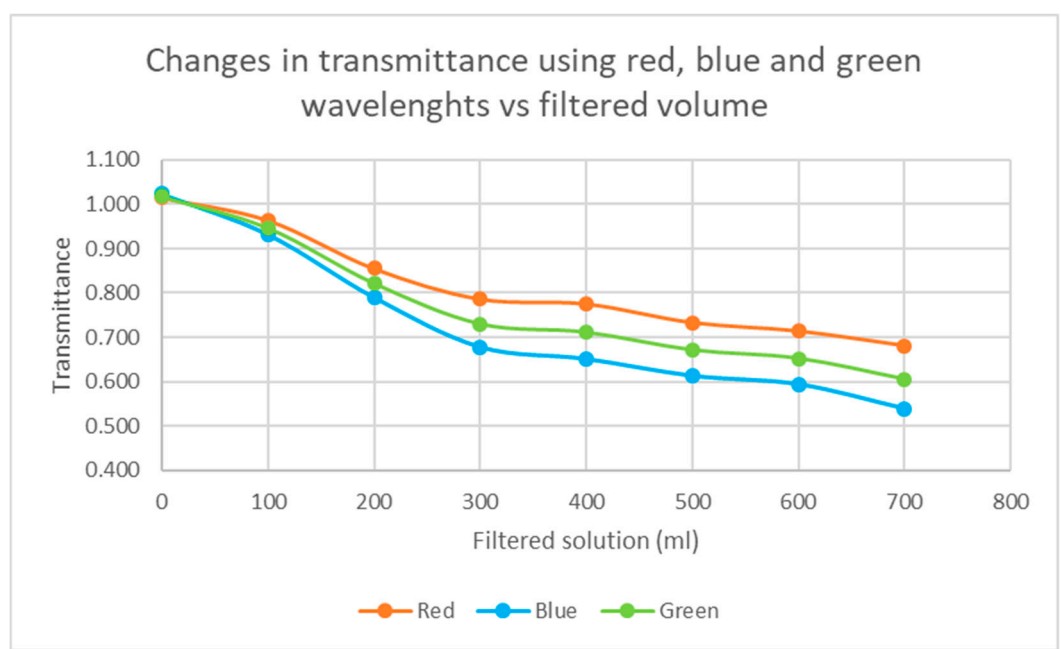

**Figure 4.** Results of the filter wetting test using a particle soot absorption photometer PSAP. Black carbon containing solution was made from wood-burning soot and ultrapure Milli-Q water.

### 3.5. The "Wrong-Side-Up Filter" Tests

The "wrong-side-up filter" tests included the following cases: the sample filter was filtered with the right or wrong side up; the sample filter was measured with PSAP with the wrong or right side up; the reference filter was measured with PSAP with the wrong or right side up. When the filter was placed with the wrong or right (hairy) side up, it was found that filtering the wrong side up does not capture BC as well as when the right side is up. In Figure 5, in the case of REF, one piece from the filter was measured against another piece of the same filter. Therefore, for correctly placed filters

(right side up, noted in the figure as R, R), any values different from 1 (e.g., >1) refer to measurement uncertainty, due to non-homogenous filter material or uncertainty related to the measurement. Here, the transmittance measurement uncertainty was found wavelength-dependent, but always <2%.

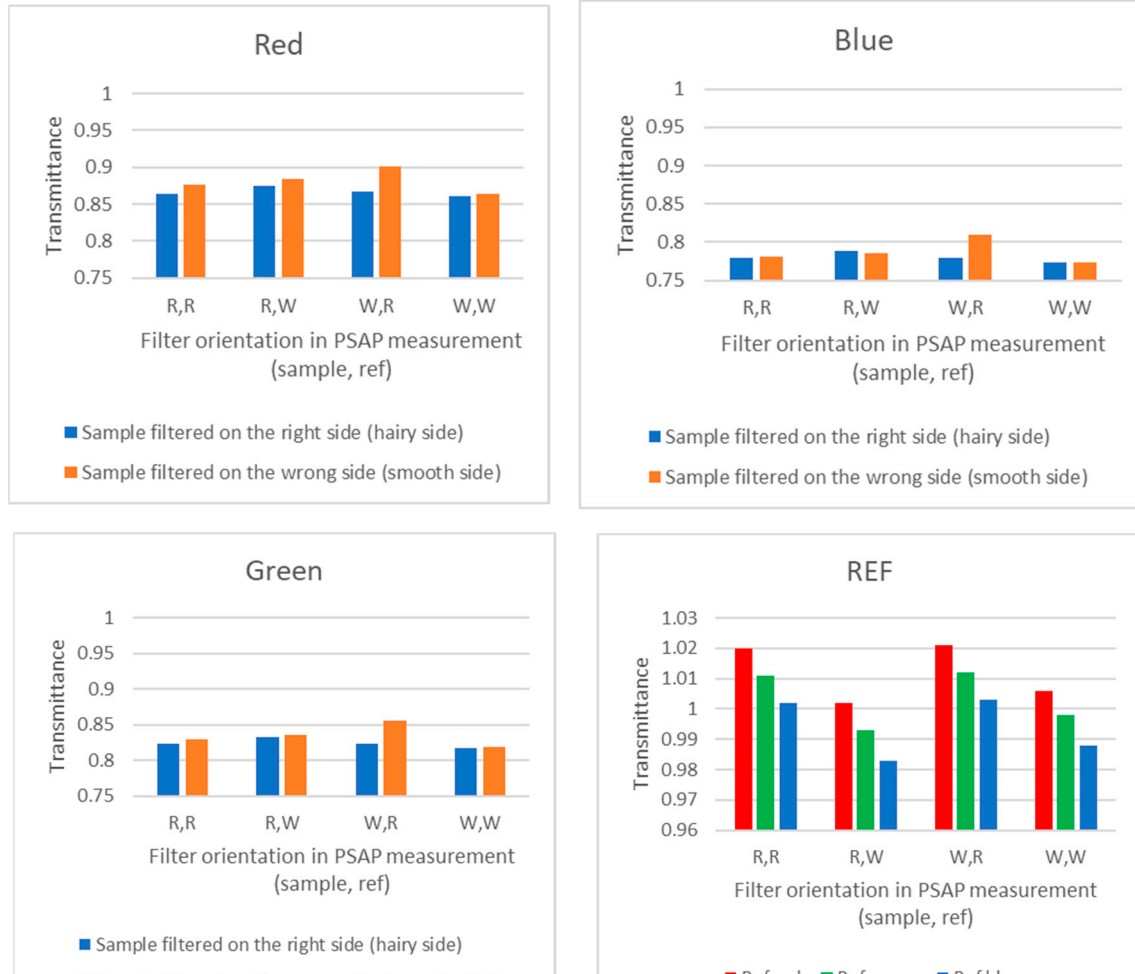

**Figure 5.** Results from the "wrong-side-up filter" test using a particle soot absorption photometer PSAP.

## 4. Discussion

The median BC, OC, and TC in snow samples ($n_{tot}$ = 30) were in Kumpula 1118, 5279, and 6396 ppb, and in Sodankylä 19, 1751, and 629 ppb. For this study, Sodankylä snow samples were collected close to a long-term (2009–to present) snow impurity site of the Sodankylä station, reported, e.g., in [6,8], as well as outside the experimental field of the SoS campaign, reported, e.g., in [7,9–11]. The values of the Sodankylä samples agree with previous studies, while for urban Kumpula, these were the first results showing carbon in the snow. Sodankylä snow impurity results for 2009–2011 suggested an increase in BC toward spring, with some variability from day to day, with concentrations between 9 and 106 ppb and some increase in OC toward the late spring (many days with >2000 ppb in April) [6]. Surface snow BC in Sodankylä during snow time in 2009–2013 was ~20 ppb during the accumulation period, increasing up to almost three times during melt [8]. Sodankylä seasonal surface snow BC concentrations in [8] were temporally highly variable. This variability was due to both atmospheric (mainly deposition of long-range transported BC) and post-depositional cryosphere processes, especially seasonal melt. The same magnitude of BC, with an average of 40 ppb, was reported in [11] for Pallas in March–April 2015 and March 2016. Black carbon values measured elsewhere in

Arctic Scandinavia (Tromsø, Norway) were 20 ppb in April, although the surface concentration can increase to 60 ppb in late May due to concentration during melting [5].

In our data, organic carbon in Sodankylä may include the organic aerosols, as well as needles and any tree trash deposited in the snow and the effect of algae and micro-organisms in the snow (Figure 6). In the boreal forest zone, OC could therefore be used as a measure of OC affecting the surface roughness of snow, which again impacts albedo. In Kumpula, organic carbon in snow is expected to consist of atmospheric aerosols and to not contain algae or tree trash, but this still needs to be confirmed.

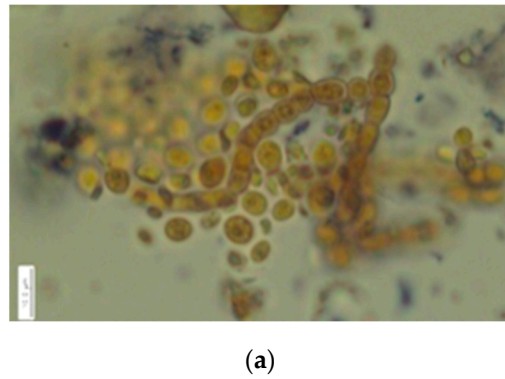
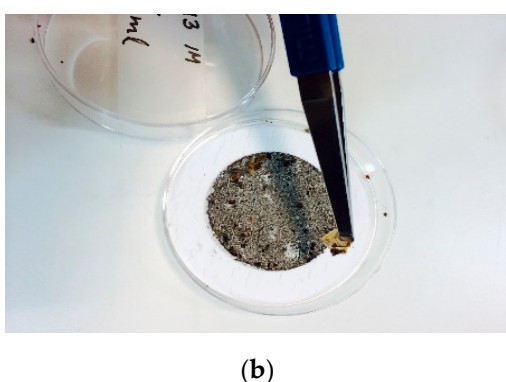

(**a**)　　　　　　　　　　　　　　　　　　　　　　　　　　　　　(**b**)

**Figure 6.** The OC content in Sodankylä snow samples has been found to be partly due to (**a**) algae (photo by Anke Kremp, SYKE, edited by Outi Meinander) pollens and other organisms in the snow, as well as (**b**) trash like needles on the filter (photo by Outi Meinander) [30].

The appearance of algae and micro-organisms in the spring snow cover have the potential to contribute to snow darkening and melt, even in amounts nonvisible to the human eye, but detectable using the OCEC method. Algae and other micro-organisms in the Sodankylä surface snow can originate from atmospheric deposition, or they can be windblown from the surrounding nature and environment, and some may reside in snow and ice and grow in the melting snow due to photosynthesis. At high latitudes, each year when winter turns into spring, solar irradiance starts to increase and is further enhanced due to multiple scattering taking place due to snow still present on the ground. In [31], it was shown that increases in the maximum daily air temperatures were not enough to initiate snowmelt in Sodankylä in the 2007 data. Instead, the importance of solar irradiance as the starting force for snow melt was discovered then. In the future, Arctic climate change may influence the environmental conditions to favor more algae and micro-organisms, up to visible amounts or resulting in color snow, previously not reported for Finland.

Here, four different types of laboratory experiment were performed in order to quantify some of the error sources. In the experiments, the following were found:

(i) Rinsing experiment: When comparing rinsing the sample bag versus not rinsing it, it was found that most of the rinsing samples were close to zero. Two samples had detectable amounts of elemental carbon, another was a sterile bag, and another was a non-sterile bag. This indicates that rinsing the sampling bag often has no effect on the detected carbon amount, but it is possible that some carbon can remain in the sample bag if not rinsed. However, the effect on the result was quite minimal, <0.01%, when compared to the carbon detected in the sample. Therefore, our findings do not suggest a need to rinse the sampling bag.

(ii) Sonication experiment: The standard deviations of the sonication cases were high due to non-homogenous snow samples and the small number of samples per case, i.e., either one or two pairs of sonicated/non-sonicated snow samples. In addition, Kumpula samples contained higher EC concentrations; the differences were therefore higher, too. In these samples, the effect of sonication was smaller than the variability due to non-homogenous snow. However, in the case of both pairs of the samples of KO and TOC (a total of four cases), the sonicated EC was higher in sonicated than

in the non-sonicated, suggesting a possible sample loss to the glass walls of the melt container. The sonication suggested a slight increase in the measured EC amount. This indicates a possibility of using sonication to extract EC attached to the container walls or to release BC from larger aggregates, which otherwise have the potential to prevent the oxidation of BC in them during the OCEC analysis. This difference, however, is small compared to the difference caused by the snow heterogeneity and the uncertainties related to the OCEC measurement. Therefore, our findings do not suggest a need to sonicate the sample container as part of the sample preparation.

(iii) Wetting experiment: When different volumes of laboratory-prepared soot-solution were filtered through quartz filters and transmittance was measured using three visible wavelengths against a blank filter, it was found that the transmittance of the filters decreased quite rapidly until a solution amount of 300 mL. After that, the decrease in transmittance was linear but slower, which might indicate that some of the soot would not stay in the filter but rather that the filter would start leaking. Further studies are needed to confirm this preliminary finding. Leaking can be corrected in multiple ways, e.g., filtering the filtration to another filter and combining these results to gain carbon from the sample.

(iv) Wrong-side-up experiment: When the filter is placed with the wrong or right (hairy) side up, it was found that filtering with the wrong side up did not capture BC as well as when the right side was up. This means that there is an underestimation of BC in the result. When using such a wrong-side-up filter for optical absorption/transmittance measurements, the error was most pronounced when putting the reference filter right side up.

To minimize sample loss, an ultrasound bath was used before filtering to remove the surface carbon from the containers. Ultrasound can also release BC from larger aggregates, which otherwise have the potential to prevent the oxidation of BC in them during the OCEC analysis. For example, results from [32] indicate that, in OCEC analysis, elemental carbon and dust often mix as agglomerates, which contributes to an underestimation of EC and OC depending on the analysis protocol. The NIOSH 5040 protocol has been argued to underestimate EC content by a factor of two [33,34]. Three different protocols (NIOSH, IMPROVE, and EUSAAR2) for thermal-optical analysis of EC and OC were compared in [35], showing good agreement (2–3%) in TC and some discrepancies (up to 40%) in EC. In turn, [5] measured mass concentrations of BC in snow ($C_{MBC}$) in Arctic regions using the integrating sphere/integrating sandwich spectrometer (ISSW) method. There, BC is estimated from spectrally resolved measurements of the absorption coefficient of solid particles collected on a filter by assuming a unitary absorption Ångstrom exponent for BC and associating most long-wavelength (650–700 nm) absorption with BC. These uncertainties are mainly due to interference from coexisting absorbing non-BC particles, such as mineral dust.

We presented in detail our current procedures and protocols to detect carbon in seasonal snow with a focus on urban backgrounds and Arctic Finland. When the effect of environmental conditions is taken into account, our protocols can be applied to environments other than those presented here. For example, when sampling in the Antarctic or the high Arctic, it may be necessary to use a clean overall cloth.

We found that the benefits and drawbacks of the OCEC method are various. OCEC is especially usable to detect both BC and OC from the same sample. There are other methods to detect carbon in snow, including SP2, optical methods such as ISSW and PSAP, and gravimetry. Based on our experience, each method has its own benefits and drawbacks, which are identified in Table 3. The critical question in choosing the method is where and why: i.e., what is the purpose of the study and what are the practical limitations? In addition to the carbon detection method, the ancillary snow data requirement also strongly depends on the planned usage of the carbon content data.

**Table 3.** Benefits and drawbacks of various methods for studies on black carbon (BC) and organic carbon (OC) in snow, based on our experience of using OCEC, SP2, optical methods, and gravimetry.

| Method | Measures | Benefits | Drawbacks |
|---|---|---|---|
| OCEC | OC, EC, TC | Detects BC, OC, and TC. TC is correct (=OC + EC). Only a small filter piece sample is used for analysis; the rest of the filter remains. European standard for atmospheric EC. Easy to filter during field work conditions. Easy to transport filters to laboratory | OC/EC ratio may depend on the protocol, as EC is based on determination of the split point. The sample is destroyed. |
| SP2 | rBC | Counts single particles. Traceable to rBC mass (ambient BC and calibration standard). It is not interfered by other compounds. | Possible breakdown of particles if the sample is pre-treated with ultrasound. Difficult to transport snow to the analysis. Destroys the sample. Data analysis is a challenge. |
| Optical methods (ISSW, PSAP) | eBC | The sample is not destroyed. Accurate if only BC is present and no other absorbing impurities. | Influence of other absorbing particles can be difficult to separate, as the measured absorption is transformed to BC using constants that may vary according to the place/chemical compounds of the sample. |
| Gravimetry | Total load | The more accurate, the larger the load. The sample is retained. | Not accurate enough for cleaner snow. Cannot separate BC from dust. |

To reveal the origin of the impurities in snow using long-range transport modeling, e.g., as used in [8], the sample collection area (e.g., one square meter), the exact time of the sample collection, and whether the sampling was within 24 h of the latest snowfall need to be known. For studies on melting snow, it is necessary to know whether impurities disturb the water-holding capacity of melting snow [7] and to have information on surface density according to the sampled snow layer. For radiative transfer RT-modeling, snow grain size, snowpack thickness/depth, and snowpack density (snowball test) are most often needed. If RTmodeling is used to verify albedo measurement, knowledge on cloudiness is essential, too (clear or fully cloudy are the best cloud conditions for this purpose).

## 5. Conclusions

We conclude that the aim of sampling in detecting carbon in snow is to measure atmospheric carbon deposition in time or space (or both), and to understand post-depositional cryospheric processes. Time can be from seconds to seasons or long-term monitoring over years or decades. Space includes spatial variability, e.g., [36], and sampling horizontally (e.g., sampling every 10 m in rows, grid, or randomly) and vertically (e.g., surface, layers, bulk, or fixed centimeters). The work benefits from including measurements on ancillary information. There, the environmental parameters relevant for carbon in snow studies depend mostly on the aim of the study. The ancillary measurements include snow depth (to separate snow accumulation), melt periods (to detect the impurity surface enrichment), snow albedo (to detect radiative effects), and precipitation (to detect the origin of the impurities), as presented, e.g., in [8]. Quality assurance (QA) and quality control (QC) as well as the identification and quantification of the uncertainties (where repetition of the measurement will produce a result within an interval around the measured value) and of the possible error sources (disagreement between a measurement and the true/accepted value) of the measured carbon data should be included in the carbon detection results. Finally, the data should be reported correctly. The distribution of BC in snow values were found to be positively skewed (number of samples $n = 107$, skewness $\gamma_1 = 0.12$), and carbon results were thus reported as median values [8]. Therefore, we recommend reporting carbon

in snow content using median values instead of average and standard deviation, which are valid for normally distributed data.

**Supplementary Materials:** The following are available online at http://www.mdpi.com/2073-4433/11/9/923/s1. Figures S1–S10; Tables S1–S2.

**Author Contributions:** Writing—original draft preparation, O.M. and E.H.; writing—review and editing, M.A. and A.H.; funding acquisition, A.H. and O.M.; supervision, O.M. and M.A. All authors have read and agreed to the published version of the manuscript.

**Funding:** This research was funded by the Academy of Finland, grant number 296302 (NABCEA) and 315497 (SnowAPP).

**Acknowledgments:** We gratefully acknowledge administrative and technical support from Terhikki Manninen/ SNORTEX-experiment and Roberta Pirazzini/SnowAPP-campaign; Aki Virkkula/PSAP; Jonas Svensson/OCEC and PSAP; the Academy of Finland A4-project (No. 254195); the Ministry for Foreign Affairs of Finland's IBA project, Black Carbon in the Eurasian Arctic and Significance Compared to Dust Sources (No. PC0TQ4BT-25); Pan-Eurasian Experiment PEEX; EU-Interact-BLACK-project, Black Carbon in Snow and Water (H2020 Grant Agreement No. 730938); EU ESSEM COST Action ES1404 (HARMOSNOW); the NordSnowNet project of the Nordregio and the Nordic Council of Ministers Arctic Co-operation Programme. OM's Ph.D. dissertation (Meinander 2016), *Effect of black carbon and Icelandic dust on snow albedo, melt and density*, University of Helsinki, is acknowledged and cited accordingly.

**Conflicts of Interest:** The authors declare no conflict of interest.

## List of Abbreviations

| | |
|---|---|
| BC | Black Carbon |
| OC | Organic Carbon |
| TC | Total Carbon |
| OCEC analysis | Thermal Optical Carbon Analysis |
| EC | Elemental Carbon |
| HULIS | Humic Like Substances |
| WMO | World Meteorological Organization |
| SNORTEX | "Snow Reflectance Transition Experiment", Sodankylä |
| SnowAPP | "Modelling of the Snow microphysical-radiative interaction and its APPlications" campaign, Sodankylä |
| GPS | Global Positioning System |
| MilliQ-water | Ultra-purified water |
| ppb | Parts per billion |
| $CO_2$ | Carbon Dioxide |
| FID | Flame Ionization Detector |
| EUSAAR2 | A protocol used in thermal optical carbon analysis (EUropean Supersites for Atmospheric Aerosol Research) |
| NIOSH 870 | A protocol used in thermal optical carbon analysis (National Institute of Occupational Safety and Health) |
| IMPROVE, IMPROVE A | A protocol used in thermal optical carbon analysis (Interagency Monitoring of Protected Visual Environments) |
| PSAP | Particle Soot Absorption Photometer |
| I | Light intensity |
| $I_0$ | Initial light intensity |
| $\tau$ | Optical depth |
| REF | Reference |
| SoS | "Soot on Snow" campaign, Sodankylä |
| $C_{MBC}$ | Black carbon mass concentration |
| ISSW | Integrating Sphere/Integrating Sandwich Spectrometer |
| SP2 | Single Particle Soot Photometer |
| RT-modeling | Radiative Transfer -modeling |
| rBC | Refractive Black Carbon |
| eBC | Equivalent Black Carbon |

**Measurement places**

| | |
|---|---|
| FMI | Finnish Meteorological Institute |
| SMEARIII | Urban background station (Helsinki, Finland) |
| GAW | Global Atmospheric Watch station (Sodankylä-Pallas, Finland) |
| KO | Kommattivaara (Sodankylä, Finland) |
| TOC | Impurities on Snow experiment, FMI, Kumpula |
| NorSEN | Nordkalotten Satellite Evaluation co-operation Network NorSEN-mast area, FMI, Sodankylä |

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
