# Peer review of "Sampling, Filtering, and Analysis Protocols to Detect Black Carbon, Organic Carbon, and Total Carbon in Seasonal Surface Snow in an Urban Background and Arctic Finland (>60° N)"

_atmosphere, doi:10.3390/atmos11090923_

Round 1

Reviewer 1 Report

Thank you for all your corrections; everything is OK now.

With best regards.

Author Response

Dear Reviewer 1,

Thank you for your acceptance.

Espoo 19 Aug 2020

Sincerely,

Outi Meinander

On the behalf of the authors

Reviewer 2 Report

I agreed with author's revision and support this publication.

Author Response

Dear Reviewer 2,

Thank you for your acceptance.

Espoo, 19 Aug 2020

Sincerely,

Outi Meinander

On the behalf of the authors

Reviewer 3 Report

The revised version of “Sampling, filtering and analysis protocols to detect Black Carbon, Organic Carbon and Total Carbon in seasonal surface snow in urban background and Arctic Finland (> 60oN)” is much better than previous. However, it will need some more minor corrections.

  1. Still it is a problem with using therm error examples: “The analytical error of the OCEC method…..” and “Errors due to carbon……” should be changed to uncertainty.
  2. Not all concentration units are unified see “Pallasbag3 (4.71 μgC) and….” etc
  3. +/-0.2 ugC + 5%. what is this?
  4. Some text editing is still necessary: FID detector is a redundancy, CO2, Figure 11. Figure 13 etc. are wrongly assigned.

A little bigger problem is still a presentation of results. Authors claim 38 measurements but Table 4 shows only 21 at most, if not 14. What happened to the rest of results? Is TC a separately measured  parameter or is only a sum of OC+EC?

After those really minor corrections this paper is ready to be published.

Author Response

This manuscript is a resubmission of an earlier submission. The following is a list of the peer review reports and author responses from that submission.

Round 1

Reviewer 1 Report

Title:

Sampling, filtering and analysis protocols to detect Black Carbon, Organic Carbon and Total Carbon in seasonal surface snow in urban background and Arctic Finland (> 60 oN)

I have the following comments:

1)

Page 5, line 141, in Figure caption

" liquid through h the filter. "

delete the " h "

---------------------

2)

Page 8, line 201, Equation 1

please correct the equation

0 should be as subscript

and exponent τ as superscript

----------------------

3) page 13, line 283, Table 4

Your values of standard deviation are quite high, is there any reason for that? Explain.

------------------------------

4) page 15, line 300, in Figure 15

Do you have any explanation why in Figure 15 REF the transmittance values are higher as 1 ? While the maximum transmittance can be 1 that mean basically 100% transmittance.

---------------------------

5) explain in the text what you mean by BC and OC. What is BC and OC? Explain the differences. Which technique you can use to distinguish these two.

----------------------

6) Specify which wavelength have you used for Red, Green and Blue filter. Specify these filters and the transmittance wavelength of these filters if possible. Explain in the text why have you decided for these three filters? Is there any particular reason for this selection?

---------------------------------

7)

At the end of the manuscript include list of all abbreviations used in the text.

Reviewer 2 Report

As it is follows from the discussion in the Sodankylä area the long-term monitoring (since 2009 - about 10 years) of snow has been conducted, so it would be interested for readers to know a few words about possible long-term trends of BC in snow cover. Although probably those estimations were already considered in some papers of the reference list.

It was found that the organic carbon in Sodankylä may include some "algae and micro-organisms" which amount increases to a spring time. It is interesting:  do those algae and micro-organisms deposit from the atmosphere or may be they appear and grow in the melting snow (due to photosynthesis)? May be authors know something about that.

The article is devoted to two important issues: 1) investigation of methodological errors in determining the concentrations of organic and mineral carbon in snow water samples and 2) quantitative assessment of the accumulation of  both OC and BC in snow cover in urban conditions and background forest conditions.

In scientific literature there are many publications on the BC, but the novelty of the authors ' results involves a detailed study of possible methodological errors in test-preparation, and for the first time a joint analysis of the content of OC and BC in same samples. In particular, were obtained interesting data on differences in the content of OC in urban and rural conditions. Of particular interest are data on the increase in the spring snow water of rural site some special OC components - algae and microorganisms.

The article is written by good scientific language and is easy to read.

The conclusion corresponds to the issues discussed in the article and answers them quite clearly.

Minor remarks .

Scientific interest in the article could be increased if the authors raised the following two issues in the discussion of the results:

  1. What can be associated with the appearance of algae and microorganisms in the spring snow cover: with precipitation from the atmosphere or with photosynthesis in snow water (usually it is rich with nitrogen compounds – as biogenic ones). It would be interesting to know the opinion of the authors.
  2. As follows from the article, the observation of the sun in the snow cover has been conducted since 2009, perhaps there are already some ideas about long-term trends in the content of sun in the snow of the studied areas. The authors could raise this issue in the discussion

Reviewer 3 Report

The manuscript: Sampling, filtering and analysis protocols to detect Black Carbon, Organic Carbon and Total Carbon in seasonal surface snow in urban background and Arctic Finland (> 60 oN) describes the analytical procedure for determination important properties of snow contaminated by mostly anthropogenic pollution.

Unfortunately this paper is written badly both in terms of English language and data presentation. This work cannot be published in the present form. However, it contains some interesting data. Therefore I will allow it for publication after full reorganization of text.

Here are some most comments for authors:

  1. You need to rethink the paper again, and present only really important findings. Text is written with very unnecessary detail. All the picture should be removed. There is no need to show biker in the ultrasonic bath! !
  2. Several parts of text are repeated. For instance see text between lines 151-158 and text between lines 161-168. Both fragments should be combined.
  3. The concentration units through the entire paper should be unify.
  4. Table 2 and 3 should be removed. Describe information from these tables in 2 sentences.
  5. Results are hard to understand. (see table 4). Present all 38 measurements and show mean values with uncertainty. Compare your measurements with the literature if it is possible.
  6. You have to make a difference between “error” and “uncertainty”.
  7. The ‘filter wrong side up’ tests do not have sense. We do experiments right, no wrong!
  8. Table 5 should go to the discussion. Some of information presented there are not the results of your experiment.
  9. Text is written in very poor English.

The above are the fundamental change you need to introduce to your text to allow me for publishing it in MDPI.
